# Socioeconomic Impacts of LCD-Treated Drinking Water Distribution in an Urban Community of the Kathmandu Valley, Nepal

**Khadga Bahadur Shrestha [1,\*]**, **Tatsuru Kamei [2]**, **Sadhana Shrestha [3]**, **Yoko Aihara [4]**,
**Arun Prasad Bhattarai [5]**, **Niranjan Bista [5]**, **Bhesh Raj Thapa [6]**, **Futaba Kazama [3]** and
**Junko Shindo [3]**

[1]  Integrated Graduate School of Medicine, Engineering and Agricultural Sciences, University of Yamanashi, 4-4-37 Takeda, Kofu, Yamanashi 400-8510, Japan
[2]  Kitasato University, 1-15-1 Kitasato, Minami-ku, Sagamihara, Kanagawa 252-0373, Japan
[3]  Interdisciplinary Centre for River Basin Environment, University of Yamanashi, 4-3-11 Takeda, Kofu, Yamanashi 400-8511, Japan
[4]  Kobe Gakuin University, 518 Arise Ikawadanicho, Nishi-ku, Kobe, Hyogo 651-2180, Japan
[5]  The Small Earth Nepal (SEN), Tripureshwor, GPO Box-20533, Tripura Marg 320, Kathmandu 44600, Nepal
[6]  International Water Management Institute (IWMI), Nepal Office, GPO Box-8975, Kathmandu EPC 416, Nepal
\*  Correspondence: khadgab@gmail.com; Tel.: +81-80-8864-2915

**Abstract:** Groundwater available in the Kathmandu Valley is not suitable for drinking due to chemical and microbial contamination. We installed a treatment system, which was made with locally available materials and was low-cost, and supplied drinking water to the intervention site where groundwater contains high amounts of ammonia, iron, and turbidity. This research aims to evaluate the socioeconomic impact of treated water distribution. One hundred households were randomly selected and asked to use treated water for drinking, and another 100 households in the nearby community were taken randomly as a control. We conducted questionnaire surveys with the enrolled households before and five months after starting water distribution to assess the water use patterns and quality perceptions. The socioeconomic impact of the intervention was evaluated by a pre-post comparison and by the difference-in-difference method. The intervention significantly enhanced most of the parameters of water quality perception, reduced the in-house water treatment, and improved the perceived water stress and quality of life. For the control site, these parameters generally became worse in the post-survey, which suggests that the survey might have affected people's mindset regarding water security. The system is an option for sustainable management of drinking water in the water-scarce, hard-hit areas in the developing countries.

**Keywords:** water treatment system; impact assessment; socioeconomic impact assessment; before and after concurrent control (BAC) design; community managed water supply system

---

## 1. Introduction

Water security is one of the greatest challenges of the 21st century due to climate change, population growth, and changing lifestyles [1,2]. Two thirds (4 billion) of the world's population experienced a water scarcity situation at least 1 month of the year [3]. The UN World Water Development Report (2015) projected that the water demand for the year 2050 would increase by 55% [4] and two thirds of the population could be under water stress conditions. The situation of the South Asia region (home to nearly 1.6 billion individuals) is clearly described by the saying *"water, water, everywhere, nor any drop to drink"* which describes a problem of scarcity amid abundance [5]. Many South Asian cities such as

New Delhi, Karachi, and Kathmandu have had water scarcity problems for many years. The water scarcity has social and economic dimensions and broadly effects on the health, nutrition, education, poverty, the environment, and society in general [6–9].

The latest census report (2011) of Nepal showed the water scarcity situation that 20% of the households had no access to the water sources on their premises and two thirds of urban households lived with an inadequate water supply [10]. The access to piped water had declined from 68% to 58% from 2003 to 2014 [11], and the Kathmandu Valley reported grossly inadequate and unreliable water supply services, where 4 million residents lived with 4.7% annual growth [12]. The Kathmandu Upatyaka Khanepani Limited has sole authority to distribute water in the urban areas of the Kathmandu Valley, but supplies only 19% of the total water demand during the dry season and 31% in the wet season in the service areas [13]. Aihara et al. (2015) showed the daily averaged water collection from the piped water was 15.4 LPCD (liters per capita per day), as lowest as <4 h/week supply [14].

The intermittent public supply and the persistent water scarcity situation of the Kathmandu Valley forced many households to use multiple water sources and water storage practices. The unmet demand of water was fulfilled by groundwater (i.e., through stone spouts and public or private wells) as a cheap, easy source, or commercial water (i.e., tanker water and jar water) as an expensive source. Groundwater sources fulfilled the water demand of 60–70% of the total water supply in the dry season, and nearly half of the wet season [15]. A rapid report on household water use in Kathmandu Valley in 2016 demonstrated that 52% of households owned private well, and 34% households relied only on the alternate sources other than the piped water [16].

Water scarcity is a substantial threat to public health and increases the risk of health and safety, stress, insecurity, poor attendance in school, time for earning, leisure, and recreation, and affects overall quality of life (QoL) [9,17,18]. In the Kathmandu Valley, many household's women and children spent much time (2–8 h/day) collecting water needed for their household [19]. The hours spent in water collection could be used in other productive ways such as earning (job), education, care for children, and other household chores. Aihara et al. (2016) found that more than 60% of postnatal mothers in the Kathmandu Valley often or sometimes worried about not having a sufficient amount of water, used less water, and had difficulty maintaining hygiene and sanitation [20]. Shrestha et al. (2018) also showed that higher water insecurity perception increased the water treatment practice and water treatment cost in the Kathmandu Valley [21]. The economic losses such as wage loss due to sickness, and costs associated with water treatment for health reasons was found highly associated with water insecurity situation [22].

Households select the best quality of water for drinking, and the decision is affected by the accessibility, reliability, perception, and cost of water. In the Kathmandu Valley, the cheapest and often-used source of drinking water is piped water; however, the supply was found to be inadequate and was rarely chlorinated at an adequate level. Eighty percentage of 46 piped water samples were contaminated with bacteria (total coliform) at the consumer tap [23]. Shrestha et al. (2013) found that *E. coli* and total coliform bacteria contamination in the tap (piped) water became higher by the longer duration of storage [24].

Jar water (the most expensive commercial water) is believed to be the best source for drinking and generally used without any treatment in many households. However, researchers have demonstrated that the jar water sold in the Kathmandu Valley was contaminated with loads of enteric bacteria and other contaminants [25–27].

Regarding the quality of groundwater in the Kathmandu Valley, many studies shown it to be very poor [28]; grossly polluted and high in iron and coliform [23,29,30]; have ammonia, nitrate, and heavy metal [30–32] content beyond the limit for drinking purposes [25]. Notably, ammonia, nitrate, and heavy metals are very difficult to remove by a simple filtration process. The high concentration of ammonia and iron has harmful effects on health as well as ammonia smells pungent and iron leaves dark stains on the surface it touches and hence such groundwater is unfit for drinking, cooking and other domestic purposes.

Thus, these literatures have shown that all types of water sources used for drinking in the Kathmandu Valley are contaminated by chemicals and/or pathogenic microorganisms, and individuals have concluded that the water is not safe. Many households practiced in-house water treatment to remove the contaminants and make it safe to drink as a coping strategy. Shrestha et al., (2018) found that 76% practiced water treatment in their households before drinking, of which 53% used more than one method; a ceramic filter (64%) and boiling (60%) were the most commonly used treatment methods [21]. Pattanayak et al. (2005) found five types of coping strategies such as collecting, pumping, storing, treating, and purchasing water adopted by the households in the Kathmandu Valley [33].

Chyasal is one of the typical water-scarce urban communities in the Kathmandu Valley, relied on groundwater sources (dug-wells and stone spouts) for their daily water needs, which was contaminated with high ammonia and iron. We designed a locally-fitted, compact, and distributed (LCD) water treatment system to improve the physical quality (turbidity) and chemical quality (nitrogen) parameters of raw groundwater, and it can provide safe drinking water, which was constructed by using locally available materials (e.g., sand, gravel, pipeline, the system body etc.) as much as possible. The system was designed to be compatible to the economic situation and to conduct sustainable drinking water treatment in the Kathmandu Valley.

In the systematic reviews on the impact of water quality improvement interventions showed higher to lower effect on preventing diarrhea [34–37] and the impact on socioeconomic status such as daily life satisfaction, sense of water security, quality perception, water use practices, and other factors were not clearly mentioned, which is a crucial aspect in preventing diarrheal diseases and overall health improvement. Stevenson et al. (2016) found a decline in water insecurity perception by the water quality improvement intervention in Ethiopia, and not found enough evidence to reduce the psychological distress [38]. In this study, we aimed to evaluate the socioeconomic impacts of LCD water distribution, such as changes in perception of drinking water, sense of water security, daily water use practice, water treatment practice and so forth in the water-scarce urban community of Nepal.

## 2. Materials and Methods

### 2.1. Research Design

We used before and after studies with concurrent control (BAC) design [39], in which the pre- and post-intervention situation was assessed by a survey at the intervention and control sites. Intervention is the LCD-treated water supply, and the water was distributed to 100 households selected randomly at the intervention site.

### 2.2. Study Area

Chyasal (ward number 9) in Lalitpur Submetropolitan city was assigned as the intervention site. Chyasal is a historical place situated approximately 500 m north from the Patan Durbar Square (Patan Palace) that extends north to the Bagmati river, a polluted river, which is the largest flowing through the Kathmandu Valley. In Chyasal, the population was 13,908; there were 3484 households, and the majority were of the Newar ethnicity [10]. Chyasal has a problem of water scarcity (piped water supply 1–3 h/week), and residents rely on dug wells and stone spouts to fulfill their daily water needs for drinking and other purposes. The groundwater is heavily contaminated with ammonium-nitrogen (16 mg N/L) and iron (9 mg/L) [40], and the National Drinking Water Quality Standards (NDWQS) are 1.2 mg N/L and 0.3 mg/L, respectively [41].

To escape from the water scarcity, a local club (Gajalaxmi club) distributed locally available groundwater treated with a bio-sand filter (so called social-club water) among the 300 member households, but the water still has a high level of ammonium-nitrogen and iron contamination. To overcome this problem, we developed a simple, cost-effective LCD system to produce safe drinking water from the available groundwater and distributed the treated water among the selected club members.

Kumbheshwor, the adjoining ward of Lalitpur Submetropolitan city (Figure 1, black color) was chosen as a control site and has a similar water scarcity situation. At the control site, a local club collected and distributed drinking water from a stone spout and one deep boring (approximately 70 m deep) without any treatment.

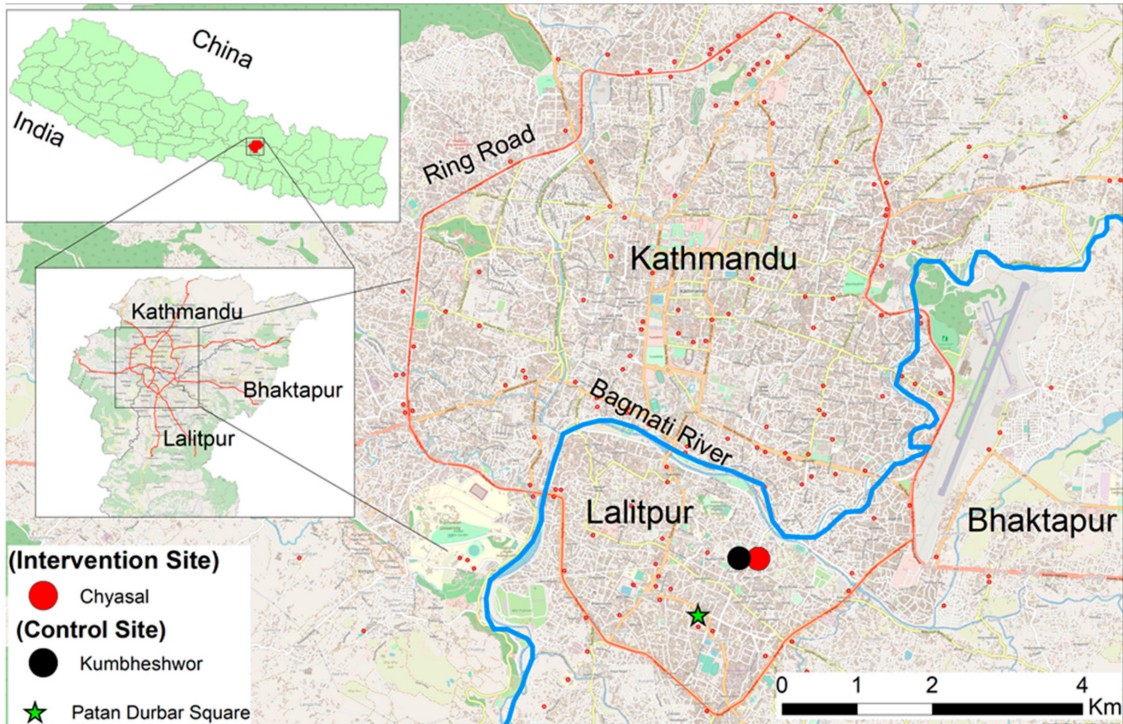

**Figure 1.** Location of intervention (Chyasal) and control (Kumbheshwor) sites in the Kathmandu Valley.

*2.3. LCD Water Treatment System*

We installed an LCD water treatment system depicted (Figure 2) comprising a sand filtration unit for iron removal, dropping nitrification process for ammonium removal, and activated carbon filtration unit for final filtration. We pumped up the groundwater into the first reservoir tank, and supplied it to the LCD systems by gravity flow at a flow rate of 1000–3000 L/day. The sand filtration unit for iron removal was constructed after the traditional filtration method used in the Kathmandu Valley. Four types of sands, namely, coarse aggregate (20–40 mm), coarse sand (>4.75 mm), medium sand (4.75–2.36 mm) and fine sand (<2.36 mm) were filled in 1000 L of tank to arrange the layers. Groundwater was supplied from the top of the reactor (down flow operation). For the ammonium removal, we made a 4-m high cylindrical reactor with a 1-m radius with locally available materials, in which, trickling filters made of acrylic fiber that are microbial carriers were suspended. Iron removed from groundwater was sprinkled into this bacterial carrier from the top of the system. Nitrifier bacteria can grow on the carrier by using ammonium in groundwater and oxygen from the air, resulting in ammonium removal by nitrification can be achieved through the system.

Next, the treated water passed through the activated charcoal and gravel filter and was stored in the final storage tank, where the required amount of chlorine solution was added and left for half an hour. Finally, the water was ready for distribution to the users.

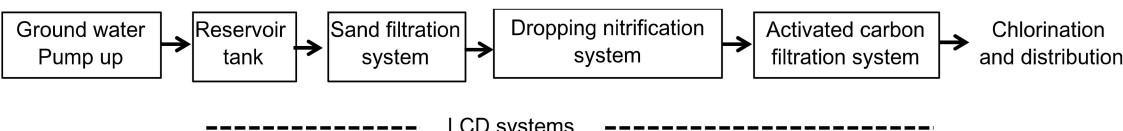

**Figure 2.** Flow diagram of the LCD (Locally-fitted, Compact, and Distributed) water treatment process.

To establish the LCD system, we had a discussion with the Gajalaxmi club committee members to learn about their perception of water quality and requirements, analyzed the water quality of available sources, and decided on the composition of the LCD system. The Gajalaxmi club operated the LCD system and distributed treated water to the enrolled households. We monitored the water quality parameters weekly in the laboratory, namely the ammonium-nitrogen, and nitrate-nitrogen concentration of the source water and LCD-treated water. The ammonium-nitrogen was analyzed by colorimetric method established by Japan Water Works Association with detection limit 0.1 mg-N/L, nitrate-nitrogen was analyzed by colorimetric method established by American Public Health Associationwith detection limit 0.5 mg-N/L. Turbidity and *E. coli* concentration were monitored time to time, not in a weekly basis by turbidity measurement instrument TU-2016, Kawasaki, Japan based on nephelometric turbidity units (NTU) and by Colilert (R) kit, respectively.

### 2.4. Intervention

We distributed 20 L of LCD-treated water per family per day as an intervention. One hundred households were randomly selected from a list of 300 members of the Gajalaxmi club in Chyasal for the LCD water distribution. We conducted a users' meeting to explain the intervention process and the roles of enrolled households. We requested that enrolled households to fetch LCD drinking water from the Gajalaxmi club during the intervention period at a fixed time with a cost of NRs 5/20 L, an amount that was decided by the club. A daily record of LCD water distribution was maintained by the focal person. LCD water distribution started in December 2017.

### 2.5. Questionnaire Survey

The semi-structured questionnaire was the main tool for data collection, and a face-to-face interview was conducted by trained interviewers from January to February, 2017 (pre-intervention), and May to June, 2018 (post-intervention). The survey was conducted with 100 households enrolled in the intervention in Chyasal. Similarly, at the control site (Kumbheshwor), 100 households were randomly selected from the list of 143 households prepared by a local club that provided relief for the Gorkha earthquake. The respondents were aged 20–60 years, were mostly housewives, and could understand and answer in the Nepali language in both communities. LCD water distribution was not informed to the respondents in the pre-intervention survey period.

The questionnaire comprised of questions on socioeconomic characteristics: including age, ethnicity, education status, occupation, family size, household amenities; water use practice: sources used, water buying, storing, and treatment practices; and perception: water quality perception, water-insecurity perception, and QoL. We selected the following socioeconomic impact indicators to compare the changes, which are described as below:

1. **Sources of drinking water used:** water source-wise percentage that households used for drinking.
2. **Water treatment practice:** percentage of households that practiced water treatment.
3. **Quality perception of main source of drinking water:** The quality perception of the main source of drinking water was measured by taste, smell, color, turbidity, and safety ranked as 1 (very poor or very unsafe); 2 (poor or unsafe); 3 (medium); 4 (good or safe); and 5 (very good or very safe). The quality perception was categorized as 1 good (very good and good) and 0 medium/poor (medium, poor and very poor) of each item.
4. **Water insecurity score (WIS):** The water insecurity was measured by 15 defined statements on the negative perception of daily water use of a 6-point rating scale ranked as 1 (never); 2 (rarely); 3 (sometimes); 4 (often); 5 (mostly); and 6 (always). The WIS was calculated by averaging the scores of all statements of every household and was regarded as a continuous variable. A higher score indicated the high insecurity perception.
5. **Quality of Life (QoL):** Questions from the World Health Organization quality of life-BREF were used to measure the QoL: 26 questions rated by 5-point scales—1 (very poor); 2 (poor); 3 (Neither

poor nor good); 4 (good); and 5 (very good). After reversing the answers of three questions (question number 3, 4, and 26) according to the manual, the QoL score was calculated by averaging all questions values and was regarded as a continuous variable. Higher scores indicate a better QoL perception.

These indicators were obtained from the pre-and post-intervention survey conducted at the intervention and control sites and compared by period. We also asked the relative quality perception of the LCD water compared with the other sources only at the intervention site:

6.  **Relative quality perception of LCD water:** Change in as quality perception of the LCD water was compared with the previous drinking source and the social-club water by the following rankings: better, same, and worse.

During the experimental period of the intervention, monitored concentrations of LCD water were not disclosed to the public to avoid the perturbation in answering the questionnaire.

Data were recorded on the questionnaire, checked by the enumerators and SEN office staffs, entered into EpiData version 3.1 (EpiData Association, Odense, Denmark), and exported into IBM SPSS Statistics version 20 (SPSS, Inc., Chicago, IL, USA) for analysis.

*2.6. Statistical Analysis*

Differences in sociodemographic and economic variables, which may have an influence on water use practice, quality perception of water, and of QoL between the intervention and control sites, was tested by a chi-square test for categorical variables and Mann–Whitney U test for ordinal variables.

To evaluate the impacts of the LCD water distribution, we conducted linear regression analysis and binary logistic regression analysis by using the Generalized Linear Models (GLM) with dummy explanatory variables—for place (1 for intervention site, and 0 for control site) and time (1 for post-intervention, and 0 for pre-intervention). The impacts were measured by DiD (difference-in-difference) that is interaction of place and time [42], which was defined as $(\overline{Y}_{\text{Intervention,post}} - \overline{Y}_{\text{Intervention,pre}}) - (\overline{Y}_{\text{Control,post}} - \overline{Y}_{\text{Control,pre}})$ for continuous variables $Y$, and as $(\text{OR}_{\text{Intervension}}/\text{OR}_{\text{Control}})$ for binary variables. Here, OR indicates the odds ratio of the odds of an event in the post-intervention period to the odds in the pre-intervention period.

*2.7. Ethical Consideration*

The study protocol of this study was reviewed and approved by the ethical review board of the University of Yamanashi (Japan) and Nepal Health Research Council with the reference application number 1 (28 November 2014) and 262/2014 (18 January 2015). The informed consent was obtained from all respondents by informing the objectives and procedures of this research before the interview. The respondents were well-informed regarding voluntary participation, withdrawing from the interview, and skipping questions they might be unwilling to answer at any time during the interview. The confidentiality and anonymity of the respondents were assured.

## 3. Results

*3.1. Quality of Source Water and LCD-Treated Water*

The result of regular water-quality monitoring showed the ammonium-nitrogen concentration was decreased sharply after intervention with the LCD treatment system. The mean concentration of source water and LCD-treated water was 11.2 ± 3.7 mg N/L and 0.6 ± 1.2 mg N/L, respectively, during the intervention period (unpublished data). The concentration of ammonia was reduced by nitrification that enhanced the nitrate concentration: the concentration of nitrate-nitrogen in the source water was 1.1 ± 0.6 mg N/L during the intervention period, but that of the treated water became 10.5 ± 1.7 mg N/L, the same level as the NDWQS limit of Nepal (11.3 mg N/L). The turbidity decreased

from 88 NTU (Nephelometric Turbidity Unit) to 0 NTU according to the occasional measurement. The source well of the LCD water contained more than $10^3$/100 mL to $10^4$/100 mL of *E. coli* bacteria, and the LCD treatment system attained approximately a 3 to 4 log reduction (Haramoto, unpublished data). Because the LCD treatment system could not always remove *E. coli* bacteria completely, we input the required amount of chlorine solution before distribution.

In the post-intervention survey, we followed up with the same households sampled in the pre-intervention survey at the intervention and control sites. We could follow up with 84 households (16 households moved out) at the intervention site, among which 78 used the LCD water as a main drinking water source. At the control site, all 100 households could be followed up.

### 3.2. Sociodemographic Characteristics of Respondents

Table 1 presents sociodemographic characteristic of respondents during the pre-intervention period.

The median age of the respondents was 40 years (median age of intervention site 39.5 years and control site 43.5 years). The ratio of the age group less than 40 years old was slightly lower at the intervention site but was not statistically significant between the intervention and control sites. The ethnicity, representing the tribal identity of the Nepalese family, was predominantly Newar, followed by Brahmin/Chhetri, Janajati, and Dalit.

Regarding respondents' highest educational attainment, the majority were educated at the school level (in primary and secondary school) or high school level and above. Notably, 16% to 17% of the respondents were illiterate. The level of education was not found to be significantly different ($p > 0.05$) between the intervention and control sites.

The majority of the respondents were unemployed and daily employed at the intervention and control sites. The respondents' occupation was not found to be significantly different at the intervention and control sites.

The economic quintile is usually obtained based on the score derived from the principal component analysis of household amenities data [43]. However, some component score coefficients became negative and scores were not considered appropriate as an economic indicator because the sample size was probably too small. We used the component score coefficient derived from the analysis using the 1500-household survey of the Kathmandu Valley conducted in 2016 [21] to calculate the score for households in this study. The *p* value showed there was not significant difference in economic quintile between the intervention and control sites.

The median number of family size was five persons (four at the intervention site, five at the control site) and ranged from 1–12 members. House ownership rate was a little higher at the control site but not significantly different, and the majority lived in their own house at both sites.

The comparison of sociodemographic at the intervention and control sites in the pre-intervention period demonstrated that all tested parameters were similar to each other, and Kumbheshwor fulfilled the condition as the control site of Chyasal in terms of the socioeconomic situation.

**Table 1.** Socioeconomic characteristics of the intervention and control sites and significance of the difference [1].

| Socioeconomic Characteristics | | Intervention (Chyasal) HH Number (%) [2] | Control (Kumbheshwor) HH Number (%) | p Value [3] |
|---|---|---|---|---|
| Age group | Less than 40 years | 33 (39.3) | 50 (50.0) | 0.146 [C] |
| | 40 years and older | 51(60.7) | 50 (50.0) | |
| Ethnicity | Brahmin/Chhetri | 4 (4.8) | 3 (3.0) | 0.309 [C] |
| | Newar | 78 (92.9) | 97 (97.0) | |
| | Janajati | 1 (1.2) | 0 (0.0) | |
| | Dalit | 1 (1.2) | 0 (0.0) | |
| Education | Illiterate | 19 (22.6) | 16 (16.0) | 0.907 [U] |
| | School level (1–10) | 33 (39.3) | 50 (50.0) | |
| | High School and above | 32 (38.1) | 34 (34.0) | |
| Occupation | Unemployed | 37 (44.0) | 38 (38.0) | 0.544 [C] |
| | Daily employed | 16 (19.0) | 25 (25.0) | |
| | Business | 14 (16.7) | 23 (23.0) | |
| | Service | 17 (20.2) | 14 (14.0) | |
| Economic quintile | Poorest | 18 (21.4) | 17 (17.0) | 0.200 [U] |
| | Poorer | 9 (10.7) | 29 (29.0) | |
| | Medium | 22 (26.2) | 15 (15.0) | |
| | Richer | 12 (14.3) | 26 (26.0) | |
| | Richest | 23 (27.4) | 13 (13.0) | |
| Family size | <5 members | 49 (58.3) | 49 (49.0) | 0.206 [C] |
| | ≥5 members | 35 (41.7) | 51 (51.0) | |
| Household ownership | Own | 77 (91.7) | 97 (97.0) | 0.110 [C] |
| | Rented | 7 (8.3) | 3 (3.0) | |

[1] Eighty-four households in Chyasal and 100 households in Kumbheshwor were used.; [2] HH is household; [3] C and U indicates a significance test was performed by using a chi-square test and by Mann–Whitney U test, respectively.

### 3.3. Drinking Water Use Practices in the Pre-Intervention and Effects of Socioeconomic Parameters

Table 2 shows the proportion of households using a particular source of drinking water, treatment practice, and treatment methods used at the intervention and control sites, and the difference between the sites and effects of the socioeconomic parameters were analyzed by the logistic regression analysis. At the intervention site, the piped water and social-club water were the common sources of drinking water, but the majority of households used the social-club water as their main source for drinking according to the survey (data were not shown). At the control site, jar water was the most commonly used source. Groundwater and tanker/vendor water were not generally used for drinking, but more than one fifth of the households used groundwater at the control site, and tanker/vendor water at the intervention site. The $\beta$ coefficient and $p$ value in the Table 2 indicated the difference between the intervention and control site, and the exponential of $\beta$ shows the adjusted odds ratio. A few socioeconomic variables have a significant association with the use of drinking water sources; social-club water, which was cheap compared with other commercial water, tended to be used at a higher rate by households in the poorest quintile, and groundwater and tanker/vender water were used at a higher rate by households with a family size with of five or more members and by the households with older housewives, respectively.

The in-house water treatment practice was common: 96% of the households at the intervention site and 85% at the control site and the difference between sites was statistically significant.

The richest, richer, and the poorer quintiles practiced water treatment at a higher rate than the poorest quintile. The ceramic filter and boiling were the most popular methods of water treatment at the intervention and control sites. The use of a ceramic filter was significantly higher in the richest

quintile than the poorest quintile, and the boiling method was practiced at a significantly higher rate by the middle groups (i.e., poorer, medium, and richer quintiles) than the poorest quintile, and by tenants compared with home owners.

**Table 2.** Situation of drinking water uses at the intervention and control sites in the pre-intervention period.

| SN | Water Use | INT | CON | $\beta$ | Sig. | Age | Edu | Occ | WQ | Fsize | HH Own |
|---|---|---|---|---|---|---|---|---|---|---|---|
| 1 | PW used for drinking | 60.7 | 5.0 | 3.691 | 0.000 | | | | | | |
| 2 | JW used for drinking | 16.7 | 61.0 | −2.244 | 0.000 | | | | | | |
| 3 | SCW used for drinking | 41.7 | 16.0 | 1.277 | 0.002 | | | | * | | |
| 4 | GW used for drinking | 9.5 | 22.0 | −1.375 | 0.009 | | | | | * | |
| 5 | TV water used for drinking | 22.6 | 4.0 | 2.342 | 0.000 | * | | | | | |
| 6 | Treat drinking water | 96.0 | 85.0 | 1.806 | 0.017 | | | | * | | |
| 7 | Ceramic filter use | 75.0 | 67.0 | 0.421 | 0.261 | | | | | | |
| 8 | Boiling | 71.4 | 55.0 | 0.919 | 0.015 | | | | ** | | * |
| 9 | Euro-guard use | 7.1 | 2.0 | 0.737 | 0.447 | | | | | | |

Note: INT = Intervention; CON = Control; PW = Piped water; JW = Jar water; SCW = Social-Club water, which is distributed by local club; GW = Groundwater (Well water, or Stone spout, or Public well); TV = Tanker or Vendor water; Edu = education; Occ= occupation; WQ = wealth quintile; Fsize = Family size; HH own = household ownership; *: significant at 5% level, **: significant at 1% level.

### 3.4. Quality Perceptions of Main Source of Drinking Water, WIS and QoL in the Pre-Intervention Period

The answers regarding the perception of quality and safety of the main source of drinking water measured by 5-point scales in pre-intervention period (Figure 3a) showed that the perception of all parameters was generally good at the intervention and control sites. Only one household at the control site perceived that the quality was poor in terms of color and turbidity. The quality perception was better at the control site: the proportion of 'very good' perception at the control site was much higher than at the intervention site, and most of the households had a positive perception including 'good' for all quality parameters and the safety at the control site.

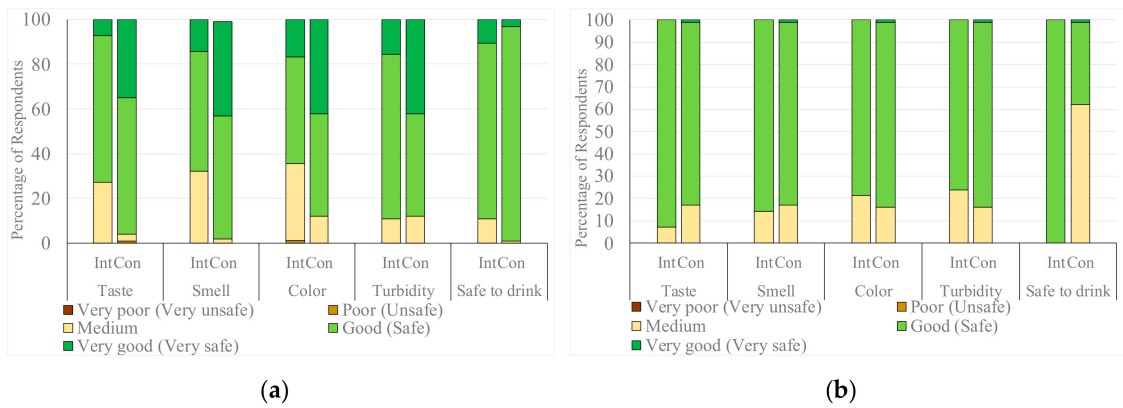

(**a**)　　　　　　　　　　　　　　　　　　　(**b**)

**Figure 3.** Perception of quality and safety of main source of drinking water in: (**a**) pre-intervention period and (**b**) post-intervention period. Int = Intervention site and Con = Control site.

WIS was directly affected by, for example, water availability and daily water management, in households, and the QoL might be affected by these water use situations indirectly. Before the analysis, we tested the internal consistency of 15 questions on water insecurity by using a reliability analysis: Cronbach's $\alpha$ for pre-intervention was 0.883, and for post-intervention was 0.887, showing good internal consistency. Similarly, the Cronbach's $\alpha$ of 26 questions on QoL was 0.856 for the pre-intervention and 0.855 for the post-intervention period. Questions on QoL were classified into four domains: physical, psychological, social, and environmental qualities. We did not observe inconsistencies between questions of different domains. The average WIS ranged from 1.1 to 3.7: the average was 2.04 ± 0.53 at the intervention site and 1.0 to 2.8 (average 1.65 ± 0.46) at the control site. The average of QoL in the intervention ranged from 1.9 to 4.0 (average 3.31 ± 0.35), and at the control

site ranged from 3.0 to 4.6 (average 3.80 ± 0.38). The $\beta$coefficients and *p* values of the regression analysis in Table 3 showed these differences were statistically significant. Socioeconomic variables namely age, education and wealth quintile showed a significant association with QoL perception, in which high school and above education had a better QoL perception than the respondents who were illiterate; the middle, richer, and the richest quintiles were a better QoL perception than poorest quintile, and the age group younger than 40 years had a better QoL perception than the age 40 years and older.

**Table 3.** Water insecurity and quality of life (QoL) perception in the pre-intervention period.

| SN | Water Use | INT | CON | $\beta$ | Sig. | Age | Edu | Occ | WQ | Fsize | HH Own |
|----|-----------|-----|-----|---------|------|-----|-----|-----|-----|-------|--------|
| 1 | Water Insecurity Score (WIS) | 2.04 | 1.65 | 0.411 | 0.000 | | | | | | |
| 2 | Quality of life | 3.31 | 3.80 | −0.509 | 0.000 | ** | ** | | ** | | |

Note: INT = Intervention; CON = Control; Edu = education; Occ= occupation; WQ = wealth quintile; Fsize = Family size; HH own = household ownership; *: significant at 5% level, **: significant at 1% level.

### 3.5. Changes in Drinking Water Sources and Quality Perceptions of the Main Source of Drinking Water in the Post-Intervention Period

The main source of drinking water was changed from social-club water and jar water to LCD water at the intervention site. Two thirds of the jar water users and all the social-club and piped water users started using the LCD water as their main source. This change showed the users' trust in the quality of the LCD water, but at the control site, jar water was the main source of drinking water in both the pre- and post-intervention period.

Figure 3b presents the proportion of each rank of drinking water perception in the post-intervention survey. The number of 'very good' and 'very safe' perception decreased markedly at both sites. At the intervention site, the proportion of 'good' increased, except for turbidity, and all the respondents perceived that the water was 'safe' to drink. At the control site, however, the proportion of positive answers decreased but was still high, except for the perception of safety. Almost all the respondents perceived safety in the pre-intervention period, and this decreased to less than half in the post-intervention period.

### 3.6. Statistical Evaluation of the Impact of LCD-Treated Water Supply

The impact of LCD water distribution was evaluated in terms of pre–post comparison and of DiD for some impact parameters (Tables 4 and 5). The quality perception of the main source of drinking water was converted into binary data and used for impact evaluation. Table 4 shows the result of binary logistic regression analysis for each binary impact parameter where effect of time, place and interaction of time and place (DiD) were presented, and Table 6 indicated the percentage of households having the positive perception of water quality or practicing water treatment. The effect of time ($\beta_t$) indicated the change in the parameter in post-period compared to that in pre-period at the control site (that was the reference class indicated by dummy variable equal to 0) and the effect of place ($\beta_p$) was the difference between intervention site and control site at the pre-intervention period (reference period). We could not get the effect of time at the intervention site in this calculation, which was shown in the last two columns that was derived by performing the same analysis by assuming the intervention site is the reference place. The model performance was evaluated by the Omnibus test in SPSS and the result for all derived models were significant at 95% of confidence level. The effect of time at the intervention site showed that the taste and smell perception were significantly improved by the pre–post comparison, the color perception was improved but not significant, and the turbidity perception was found significantly deteriorated (Tables 4 and 6). At the control site, however, all the quality parameters deteriorated, but changes in perception of color and turbidity were not significant. The difference between intervention and control sites were significant in all quality parameters except turbidity, which was poorer in the intervention site. Safety perception improved at the intervention

site, but the $\beta$ coefficient could not be calculated (Table 4) because all data were 1 (safe) in the post intervention period and decreased significantly at the control site (Table 6).

The DiD showed that perceptions of taste and smell at the intervention site were significantly improved compared with the control site. However, the color perception was improved ($\beta = 0.94$, $p > 0.05$) and the turbidity perception was deteriorated ($\beta = -0.78$, $p > 0.05$), but were not significant.

In accordance with the increasing perception of water safety by the LCD-treated water distribution, the ratio of households practicing water treatment before drinking decreased at the intervention site. This decrease was statistically significant and was highly significant based on the DiD evaluation considering the changes at the control site, where the treatment practice significantly increased (Tables 4 and 6).

Table 5 presents the result of impact evaluation for the continuous impact variables: WIS and QoL. At the intervention site, WIS in the pre- to post-intervention period decreased (average WIS became ($1.90 \pm 0.34$); thus, the perception of water insecurity improved, but the difference was small and not a significant change. However, the score significantly increased to $2.11 \pm 0.60$ at the control site, and DiD demonstrated the significant decrease compared with changes at the control site. The insecurity score was significantly lower in the service holder than the unemployed. The perceived QoL at the intervention site improved to $3.53 \pm 0.16$ that was significant based on the effect of time ($\beta_{t1}$) and by the DiD analysis ($\beta_{t.p}$). The QoL perception was significantly higher in high school and above education than the illiterate; in the higher wealth quintiles (i.e., medium, richer and richest quintiles) than the poorest quintile, however, significantly lower in the 40 years and above age group than the younger.

**Table 4.** Statistical evaluation of the impact of LCD-treated water distribution on binary impact variables [1].

| SN | Impact Variables | Time Effect ($\beta_t$) | | Place Effect ($\beta_p$) | | (DiD = $\beta_{t.p}$) | | Socioeconomic Parameters [2] | | | | Time Effect at Intervention Site ($\beta_{t1}$) | |
|---|---|---|---|---|---|---|---|---|---|---|---|---|---|
| | | Adjusted ($\beta$) Coff | $p$ Value | Adjusted ($\beta$) Coff | $p$ Value | Adjusted ($\beta$) Coff | $p$ Value | Age | Edu | Occ | WQ | Adjusted ($\beta$) Coff | $p$ Value |
| 1 | DW taste perception | −1.68 | 0.004 | −2.16 | 0.000 | 3.18 | 0.000 | | | | | 1.49 | 0.003 |
| 2 | DW smell perception | −2.43 | 0.002 | −3.19 | 0.000 | 3.39 | 0.000 | * | | | | 0.96 | 0.020 |
| 3 | DW color perception | −0.35 | 0.393 | −1.43 | 0.000 | 0.94 | 0.088 | | | | | 0.59 | 0.106 |
| 4 | DW turbidity perception | −0.50 | 0.222 | 0.38 | 0.467 | −0.78 | 0.218 | | | | | −1.27 | 0.008 |
| 5 | Safe to drink [3] | −5.41 | 0.000 | −2.40 | 0.027 | – | – | | | * | | 20.38 | 0.998 |
| 6 | DW Treatment | 0.13 | 0.003 | 0.11 | 0.020 | −0.44 | 0.000 | | | | | −0.31 | 0.000 |

Note:

1. Binary logistic regression was applied.
2. *: significant at 5% level.
3. $\beta$ coefficient of intervention site and DiD cannot be calculated because 100% of households perceived the water was safe to drink in the post-intervention.

**Table 5.** Statistical evaluation of the impact of LCD-treated water distribution on continuous impact variables [1].

| SN | Impact Variables | Time Effect ($\beta_t$) | | Place Effect ($\beta_p$) | | (DiD= $\beta_{t.p}$) | | Socioeconomic Parameters [2] | | | | Time Effect at Intervention Site ($\beta_{t1}$) | |
|---|---|---|---|---|---|---|---|---|---|---|---|---|---|
| | | Adjusted ($\beta$) Coff | $p$ Value | Adjusted ($\beta$) Coff | $p$ Value | Adjusted ($\beta$) Coff | $p$ Value | Age | Edu | Occ | WQ | Adjusted ($\beta$) Coff | $p$ Value |
| 1 | Water Insecurity Score | 0.47 | 0.000 | 0.39 | 0.000 | 0.61 | 0.000 | | | * | | −0.14 | 0.071 |
| 2 | Quality of Life Score | 0.14 | 0.071 | −0.49 | 0.000 | 0.65 | 0.000 | ** | ** | | ** | 0.22 | 0.000 |

Note:

1. Linear regression was applied

*: significant at 5% level, **: significant at 1% level.

**Table 6.** Change in water quality perception and treatment practice at intervention and control sites.

| SN | Impact Variables | Intervention Site | | | Control Site | | |
|---|---|---|---|---|---|---|---|
| | | HH no | Pre-int | Post-int | HH no | Pre-int | Post-int |
| 1 | DW taste perception | 78 | 74% | 92% | 100 | 96% | 83% |
| 2 | DW smell perception | 78 | 69% | 85% | 100 | 98% | 83% |
| 3 | DW color perception | 78 | 65% | 77% | 100 | 88% | 84% |
| 4 | DW turbidity perception | 78 | 91% | 74% | 100 | 88% | 82% |
| 5 | Safe to drink | 78 | 90% | 100% | 100 | 99% | 38% |
| 6 | DW Treatment | 84 | 96% | 65% | 100 | 85% | 98% |

Note: HH = Household; Pre-int = Pre-intervention; Post-int = Post-intervention; DW= Drinking Water.

### 3.7. Relative Quality Perception of LCD Water with Previous Main Sources and Social-Club Water

Figure 4 presents the result of the relative quality perception of LCD-treated water compared with the previous main drinking water source (Figure 4a) and with social-club water (Figure 4b) based on the answers of 78 LCD-user households. More than half of the respondents mentioned that the quality of the LCD water was better than the previous main source of drinking water, and more than 70% of respondents perceived the quality was better than the social-club water in all quality parameters. None of the respondents perceived that the LCD water was worse in all quality parameters than the previous main source and social-club water. Regarding the perception of specific quality, the taste perception improved the most in both source comparisons (71% and 80%), and the turbidity perception improved the least (53% and 71%). The effect of the intervention was evaluated based on the quality perceptions of the pre- and post-intervention periods and showed the turbidity perception deteriorated significantly, which was substantially different from the result by relative perception. Similarly, we measured respondents' attitude toward the effects of LCD water, assessed by positive and negative statements, in which more than 95% agreed the cost of buying water and the sense of insecurity regarding the drinking water decreased (data not shown).

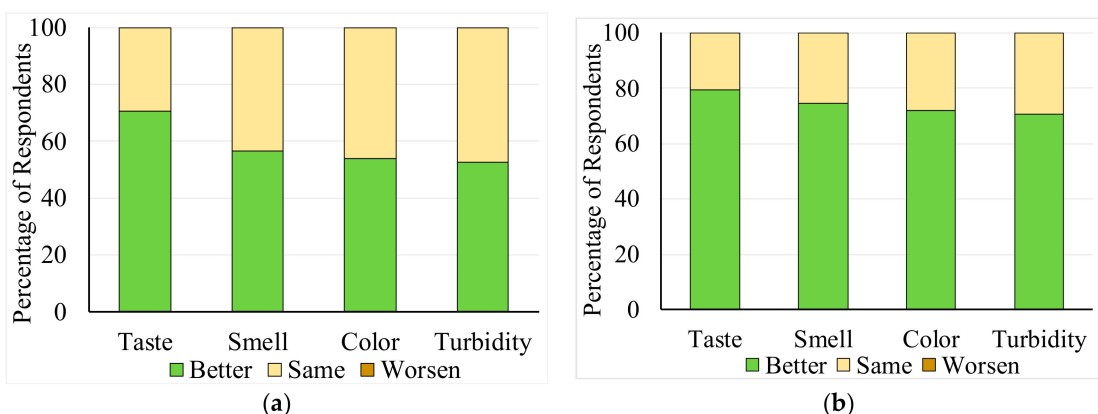

**Figure 4.** Relative quality perception of LCD water compared with: (**a**) previous main source and (**b**) social-club water.

## 4. Discussion

### 4.1. Water Quality Perception

The quality perception of water is one of the important factors in deciding the water use behavior and sustainability, for example, selection of water source, continuation of using the water source, and coping strategies for averting poor-quality water. Francis et al. (2015) concluded that taste and odor were the key factors for acceptance and sustainability of water quality intervention [44]; Rojas and Megerle (2010) found the color and appearance (turbidity) were the main factors for quality

perception [45]; and Doria (2010) highlighted the organoleptic properties, especially taste, smell, color, and turbidity, as the major factors in quality perception [46]. In this research, we compared the quality perception of drinking water based on the taste, smell, color, turbidity, and safety. The improved quality of the LCD-treated water helped to enhance the quality perception of the LCD water and as a result, more than 90% of the surveyed households at the intervention site adopted the LCD water as their main source of drinking water and reduced the use of jar water and social-club water for drinking. However, these organoleptic parameters are subjective and the criteria of judgment depend on the individual person and are changeable. Although we found the improved quality perception after the LCD water distribution, some results were inconsistent. Perception of turbidity of the LCD water was resulted to be poorer than the previous main source of drinking water according to the comparison of the pre-and post-intervention results (Figure 3a,b and Table 4), but no households perceived that the turbidity of the LCD water was worse than the previous main source of water according to the relative perception of quality change (Figure 4a). Similarly, we obtained the direction of perception change in an individual household in the water quality from the pre- to post-intervention period, and we categorized the change into improved (positive change), same (no change), and worsen (negative change) and demonstrated in Figure 5). The result showed the *worsen* perception was found in all quality parameters, not only for turbidity, which was not mentioned in the relative comparison (Figure 4a), and also the proportion of 'improved' quality perception was lower than the relative quality perception (Figure 4a).

One reason for these inconsistencies is that the results in Figure 4a,b were affected by recall bias about the situation in the pre-intervention period that was almost 1 year before the post-intervention survey. Another possible reason for the inconsistency is the changing criterion of the judgment of water quality during 1 year: more specifically, individuals might have desired higher quality and safer drinking water; thus, the perception in the post-intervention survey was stricter than before. This notion was supported by the results at the control site: all perception parameters became worse, especially for the perception of safety, and the number of households applying in-house treatment of drinking water significantly increased in the post-intervention survey. The reason for the change was unclear, but our survey might have been a trigger for changing the respondents' mindset regarding water security.

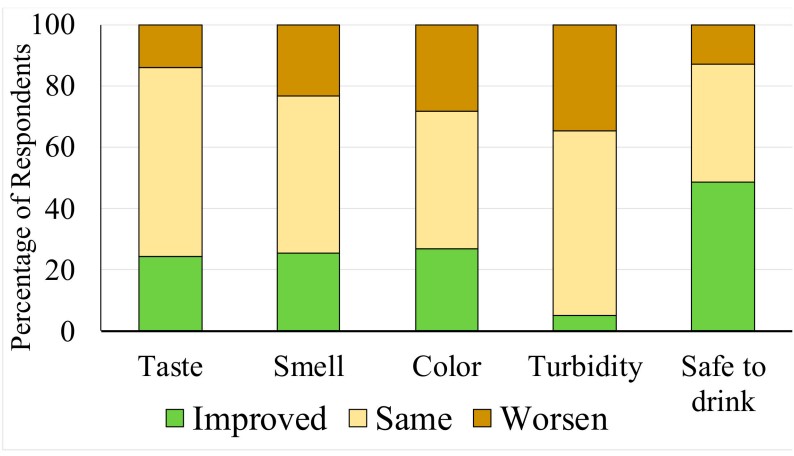

**Figure 5.** Direction of change in drinking water quality perception among LCD users.

### 4.2. In-House Water Treatment Practice and Cost

The perception of water quality and risk to health triggers the decision to use a water source, and use of in-house water treatment practices [47]. Onjala et al. (2014) found that the choice of water treatment was significantly correlated with the quality and risk perception of water [48]. Households select the safest and most reliable source of water for drinking, buy the safest source of water from the market, or practice in-house treatment, and this is triggered by the quality perception, risk to

health perception, social norms [49], and the insecurity perception [21]. In our study, the practice of in-house water treatment was significantly reduced after the use of the LCD water as their main source of drinking water at the intervention site, but 65% of households continued to practice treatment. Some households used another source of water for drinking purposes after treatment together with the LCD-treated water. Although more than 90% of the households drank the LCD water without treatment, some households applied treatment to reduce the risk to their young children and senior family members. According to the personal communication with the respondents, some households were using their ceramic filter just for only storing purposes, and this was counted as households using a treatment practice. The reduction in the cost of in-house treatment was notable due to a decrease in the number of households treating drinking water and the decreased in the per household amount of water to be treated that was difficult to be estimated from the survey data.

The decreased treatment practice could help to minimize the burden on the household expenditure, and the cost saving could be used for other purposes such as food, education, health, recreation, and other households' necessities; this is one of the positive socioeconomic impacts of LCD water distribution.

*4.3. WIS and QoL Improvement*

Water insecurity perception is one of the crucial short-term emotional stressors. The situation of an intermittent and unreliable piped water supply in the Kathmandu Valley severely constrained the accessibility, adequacy, and daily lifestyle of residents [50] and often increased the insecurity perception [51]. We measured insecurity perception based on the WIS, and at the intervention site, the effect of the daily 20-L LCD water distribution significantly reduced the WIS compared with the control site. Result in Stevenson et al. (2016) also demonstrated that the improvement in the community water supply helped to reduces the insecurity perception [38]. The water security perception improves the sense of security and reduce the stress in the community. Surprisingly, even in a short period of LCD water distribution at the intervention site, the better water security perception increased the house-rent and occupancy rate of the rented households, as mentioned by the committee members and LCD users during the field visit.

QoL, however, is the overall general well-being of an individual and family that observed from the life satisfaction, and includes physical health, education, employment, wealth, safety and security, social belongings, religious belief, and environment [52]. Water scarcity situation reduces the standard of living and QoL, especially for the vulnerable women and children, who are more vulnerable than men, by increasing the unnecessary burden of water collection and transportation. The result of our study found the QoL increased significantly at the intervention site despite the rather short intervention period. The larger effect of LCD water distribution on the QoL improvement probably indicates that water scarcity situation, anxiety regarding water safety, and the coping activity are substantial stresses in the respondents' daily life in the Kathmandu Valley. Aihara et al. (2015) conducted research in the Kathmandu Valley and showed that 11% of the respondents felt their QoL was deteriorated always or most of the time by the water scarcity problems, and the regression analysis showed a moderate inverse relation between QoL and water scarcity perception [14]. This result showed that LCD water distribution helped to improve the way of daily life and elevated QoL perception at the intervention site.

We focused on the immediate impacts of LCD-treated water distribution because of the short intervention period: 5 months from the start of the intervention to the post-intervention survey. Notably, the LCD-treated water continued to be supplied. Because of the short experimental period, we could not evaluate the long-term impacts such as changes in the household economy, nutrition status, health status, and school attendance. Our target area was one small urban community in the Kathmandu Valley, and our results were site-specific. Thus, the impact of the LCD-treated water supply might be different in other communities due to the differences in their natural and social conditions. However, our target community, Chyasal, is a typical traditional community in water-scarce areas in

the Kathmandu Valley, and our results would be helpful in the improvement of water scarcity in the similar communities. Further research should include diverse geographical areas, a larger sample size, and a longer intervention period to measure the immediate and long-term impacts over a wider area.

**5. Conclusions**

In this paper, we evaluated the socioeconomic impacts of the LCD-treated water distribution. The organoleptic parameters such as taste, smell, color, turbidity, and perception of drinking water safety, water insecurity, and QoL were used as measures for the impact evaluation. Based on the pre–post comparison and DiD considering the changes at the control site, almost all the parameters were perceived to have been improved and minimized the in-house water treatment practice and its associated treatment cost, which reduces the burden of the households: water treatment cost. This study mainly based on the users' perception that might change by time, which limits us to get the stable results. It focused only one urban community which may be different the water scarcity situation with other such area. This limits us to generalize our socioeconomic impacts of LCD-treated water distribution in whole urban population. In spite of these limitations, this intervention of water quality improvement managed by the community clearly demonstrated the positive socioeconomic impacts that enhance daily life and represents one option for sustainable management of safe drinking water distribution in the water-scarce urban communities of developing countries.

**Author Contributions:** J.S., F.K., and K.B.S. conceived and designed this research; K.B.S., J.S., S.S., F.K., and Y.A. designed the survey and prepared questionnaire; A.P.B. and N.B. implemented the surveys and entered data; K.B.S. and J.S. performed the statistical analysis and interpretation of data; K.B.S. prepared the manuscript; and J.S., F.K., T.K., B.R.T., Y.A., and S.S. developed the concept and revised the manuscript critically. All authors contributed to the final manuscript.

**Funding:** This study was conducted as a part of "Hydro-microbiological approach for water security in Kathmandu Valley" project of University of Yamanashi under the SATREPS program, jointly funded by JICA and JST.

**Acknowledgments:** We would like to appreciate all the respondents who provided their valuable perspective in the questionnaire survey, and the hard work of interviewers during data collection. We express our sincere gratitude to the Science and Technology Research Partnership for Sustainable Development Program (SATREPS) project lead by the University of Yamanashi, funded by Japan International Cooperation Agency (JICA) and Japan Science and Technology Agency (JST). We are indebted to Eiji Haramoto for providing information about *E. coli* reduction by the LCD system and Bijaya Man Shakya for making a map in ArcGIS software.

**Conflicts of Interest:** The authors declare no conflict of interests.

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
