# Peer review of "Socioeconomic Impacts of LCD-Treated Drinking Water Distribution in an Urban Community of the Kathmandu Valley, Nepal"

_water, doi:10.3390/w11071323_

Round 1
Reviewer 1 Report
The introduction provides sufficient background but can be improved including other relevant references in this issue:
For example in line 42: Documents published by World Health Organization available at https://www.who.int/water_sanitation_health/water-quality/household/en/ can be added, or other publication in this context: e.g. Valeriani, F et al. International Journal of River Basin Management 2015;13(3):325-331).
In the material and methods:
In line 173: Some water quality parameters are lacked in the list of performed analyses: e.g. turbidity, E. coli. Moreover, methods used for their detection must be added with the reference technical standards and the limits of detection.
Results:
In line 251-258: You check that the water quality parameters are reported in the material and methods.
In table 1: Check the acronyms, e.g. HH. The same for other tables and figures.
Other comments: Improvements of English language and style are required.
Author Response
We all authors are very much thankful to the reviewers for their valuable comments on our manuscript. Below we provide the responses to the comments and questions raised. Modifications and improvements are incorporated in the revised manuscript as mentioned below for each of the comments. The responses to the reviewers’ comments are provided below and changes in manuscript are in track change form for easy visualization. Some old figures and tables are replaced with new figures and tables. please find attached pdf file.

Reviewer 2 Report
This paper is not proper for this journal due to socioeconomic research which can not be repeated and due to small number of responders.
Author Response

(The authors gave the same response as above.)

Reviewer 3 Report
No comments are need.
Author Response

(The authors gave the same response as above.)

Reviewer 4 Report
Please see my review comments to improve the paper. File attached.

Author Response

(The authors gave the same response as above.)

Round 2
Reviewer 4 Report
The paper has been greatly improved and my comments&concerns have been appropriately addressed. The paper can be published in its current version.